# Effects of Multi-Bacteria Solid-State Fermented Diets with Different Crude Fiber Levels on Growth Performance, Nutrient Digestibility, and Microbial Flora of Finishing Pigs

**DOI:** 10.3390/ani11113079

**Published:** 2021-10-28

**Authors:** Ping Hu, Lingang Wang, Zhijin Hu, Liwen Jiang, Hong Hu, Zebin Rao, Liuting Wu, Zhiru Tang

**Affiliations:** Laboratory for Bio-Feed and Animal Nutrition, Chongqing Key Laboratory of Herbivore Science, Southwest University, Chongqing 400715, China; huping0913@163.com (P.H.); wlg18700217533@163.com (L.W.); hzj039@163.com (Z.H.); jlw18080401887@163.com (L.J.); huhong1020@163.com (H.H.); raozebin2020@163.com (Z.R.); 17752785935@163.com (L.W.)

**Keywords:** dietary fiber level, multi-bacteria solid-state fermentation, growth performance, digestibility, microflora

## Abstract

**Simple Summary:**

Dietary cellulase was found to be an important nutrient, and solid-state fermentation could improve the nutritional value of feed. To study the effects of multi-bacteria solid-state fermented diets and dietary crude fiber levels on finishing pigs, a total of 36 pigs were divided into four treatments: (1) pigs fed a basal diet containing 7.00% CF (HF), (2) pigs fed a basal multi-bacteria fermentation diet containing 7.00% CF (HFM), (3) pigs fed a basal diet containing 2.52% CF (LF), and (4) piglets fed a basal multi-bacteria fermentation diet containing 2.52% CF (LFM). The growth performance, nutrient digestibility and digestion amount, serum biochemical index, and fecal microflora were evaluated. Multi-bacteria solid-state fermentation had a positive effect on the nutrient digestion and serum biochemical indicators, which was contrary to high-fiber diets. Both high-fiber diets and multi-bacteria solid-state fermentation could optimize intestinal flora in finishing pigs.

**Abstract:**

This study aimed to investigate the effects of multi-bacteria solid-state fermented diets with different crude fiber (CF) levels on growth performance, nutrient digestibility, and microbial flora of finishing pigs. The multi-bacteria solid-state fermented diets were made up of *Lactobacillus amylovorus*, *Enterococcus faecalis*, *Bacillus subtilis*, and *Candida utilis*. According to a 2 (factors) × 2 (levels) design, with the two factors being multi-bacteria solid-state fermentation (fed non-fermented diet or multi-bacteria fermentation) or CF levels (fed a basal diet containing 2.52% CF or 7.00% CF), a total of 36 finishing pigs (70.80 ± 5.75 kg) were divided into 4 treatments with 9 barrows per group: (1) pigs fed a diet containing 7.00% CF (HF), (2) pigs fed a multi-bacteria fermentation diet containing 7.00% CF (HFM), (3) pigs fed a diet containing 2.52% CF (LF), and (4) piglets fed a multi-bacteria fermentation diet containing 2.52% CF (LFM). This experiment lasted 28 days. The multi-bacteria solid-state fermented diet increased the backfat thickness (*p* < 0.05) and apparent total tract nutrient digestibility (ATTD) of CF, neutral detergent fiber (NDF), acid detergent fiber (ADF), crude protein (CP), 8 amino acids (Trp, Asp, Gly, Cys, Val, Met, Ile, and Leu), total essential amino acids (EAA), total non-essential amino acids (NEEA), and total amino acids (TAA) (*p* < 0.05). Multi-bacteria solid-state fermented diet increased serum concentrations of HDL-c, ABL, TP, and GLU, the serum enzyme activities of GSH-Px, T-AOC, SOD, and CAT (*p* < 0.05), the relative abundance of *Lactobacillus*, *Oscillospira*, and *Coprococcus* (*p* < 0.05), and the abundance of YAMINSYN3-PWY, PWY-7013, GOLPDLCAT-PWY, ARGORNPROST-PWY, and PWY-5022 pathways (*p* < 0.05). The multi-bacteria solid-state fermented diet reduced the digestion amount of CF, NDF, and ADF (*p* < 0.05), the serum concentrations of TC, TG, LDL-c, BUN, and MDA (*p* < 0.05), the relative abundance of *Streptococcaceae* (*p* < 0.05), and the abundance of PWY-6470, PWY0-862, HSERMETANA-PWY, LACTOSECAT-PWY, MET-SAM-PWY, PWY-6700, PWY-5347, PWY0-1061, and LACTOSECAT-PWY pathways (*p* < 0.05). The high-fiber diet increased average daily feed intake (*p* < 0.05), the serum concentrations of TC, TG, LDL-c, BUN, and MDA (*p* < 0.05), the relative abundance of *Clostridiaceae_Clostridium* and *Coprococcus* (*p* < 0.05), and the abundance of TCA-GLYOX-BYPASS, GLYCOLYSIS-TCA-GLYOX-BYPASS, and PWY-6906 pathways (*p* < 0.05). The high-fiber diet reduced chest circumference (*p* < 0.05) and ATTD of ether extract (EE), CF, NDF, ADF, Ca, CP, 18 amino acids (Trp, Thr, Val, Met, Ile, Leu, Phe, Lys, His, Arg Asp, Ser, Glu, Gly, Ala, Cys, Tyr, and Pro), EAA, NEAA, and TAA (*p* < 0.05). The high-fiber diet also reduced the serum concentrations of HDL-c, TP, ABL, and GLU, the serum enzyme activities of T-AOC, GSH-Px, SOD, and CAT (*p* < 0.05), and the relative abundance of *Akkermansia* and *Oscillospira* (*p* < 0.05). There was no significant effect of the interaction between multi-bacteria fermentation and dietary CF levels, except on the digestion amount of CF (*p* < 0.05). The 7.00% CF had a negative effect on the digestion of nutrients, but multi-bacteria solid-state fermentation diets could relieve this negative effect and increase backfat thickness. High-fiber diets and multi-bacteria solid-state fermentation improved the diversity and abundance of fecal microorganisms in finishing pigs.

## 1. Introduction

High dietary crude fiber (CF) levels usually had a negative impact on feed utilization, such as anti-nutritional factors and the decrease of nutrient digestibility, which was related to the concentration and composition of CF, processing technology, and the animal itself [1]. However, because of a good tolerance to fibrous feed, the growth rate and nutrient digestibility of Chinese local pig breeds did not change after pigs were fed diets with 7.57% or 9.7% CF [2,3], and finishing pigs could digest and utilize CF better than piglets [4,5]. Although CF was not called a nutrient and had a certain anti-nutritional effect, it still had nutritional value, especially for hindgut health and intestinal microbes [6]. CF had the benefit of promoting intestine microbiota fermentation in animal, and feeding diets with high fiber levels could increase the relative abundance of probiotic bacteria and promote SCFAs production, which was beneficial to intestinal health and microbial balance [7,8]. Therefore, the research focus on how to avoid the negative effects of CF on nutrition and to better exert its probiotic effects.

Solid-state fermentation referred to the fermentation by microorganisms in a solid substrate with little free water. Cellulase and xylanase produced via this process could degrade CF in feed and promote the digestion and utilization of CF [9]. In addition, studies had shown that the role of probiotics had genus, species, and strain specificity, and the multi-bacteria fermentation with synergy would be more effective than a single probiotic [10,11]. For example, as aerobic bacteria, *Bacillus subtilis* and *Candida utilis* were able to consume oxygen, create an anaerobic environment for *Lactobacillus amylovorus* and *Enterococcus faecalis,* and promote their growth. Besides, *Lactobacillus amylovorus* could produce lactic acid to reduce the pH of fermented feed, inhibit the growth and reproduction of Gram-negative pathogens, reduce body serum cholesterol, ammonia nitrogen, the amount of hemicellulose and lignin in fermented feed, and increase soluble carbohydrate content [12]. *Enterococcus faecalis* could produce lactic acid and SCFAs and regulate lipid metabolism. Its metabolized hydrogen peroxide and mannan peptide could kill harmful bacteria such as *Escherichia coli* and *Salmonella* [13,14]. *Bacillus subtilis* was used to consume free oxygen, to create a favorable environment for anaerobic probiotics such as *Lactobacillus*, and to inhibit the growth of harmful bacteria to adjust the digestive tract flora [15]. *Bacillus subtilis* secreted a variety of digestive enzymes for proteins and starch, and eliminated anti-nutritional factors as well [16]. *Candida utilis* was an excellent source of amino acids, nucleic acids, vitamins, and a variety of digestive enzymes that are beneficial to animals (α-amylase, protease, cellulase, etc.), benefiting animal digestion [17]. The cell wall of *Candida utilis* contained dextran, mannan, and other active substances, which had been proven to increase the expression of Occludin and β-definsin-2, promote the development of the intestinal barrier, and enhance animal immune function [18,19].

To study multi-bacteria solid-state fermentation, improving high dietary CF levels’ adverse effect on the nutrient digestibility, the *Lactobacillus amylovorus* TZR-PI001, *Enterococcus faecalis* TZR-PI003, *Bacillus subtilis* ATCC 19659, and *Candida utilis* CICC 1314 strains were applied in solid-state fermentation diets with dietary 2.52% or 7.00% CF levels.

## 2. Materials and Methods

### 2.1. Strain, Culture Media, and Reagents

*Lactobacillus amylovorus* TZR-PI001 and *Enterococcus faecalis* TZR-PI003 were provided by the Applied Microbiology Laboratory of the College of Animal Science and Technology of Southwest University. *Bacillus subtilis* ATCC 19659 was purchased from American Type Culture Collection (ATCC). *Candida utilis* CICC 1314 was purchased from the China Center of Industrial Culture Collection (CICC).

MRS culture medium comprised 10.0 g beef extract, 10.0 g peptone, 5.0 g sodium acetate, 5.0 g yeast powder, 20.0 g glucose, 2.0 g diammonium citrate, 1.0 g Tween 80, 2.0 g K_2_HPO_4_, 0.58 g MgSO_4_·7H_2_O, 0.25 g MnSO_4_·H_2_O, 15.0 g agar, and 1.0 L distilled water. This medium was adjusted to pH 6.8 and sterilized for 15 min at 115 °C.

TSA medium comprised 15.0 g tryptone, 5.0 g peptone, 5.0 g NaCl, 13.0 g agar, and 1.0 L distilled water. This medium was adjusted to pH 7.3 and sterilized for 15 min at 115 °C.

YPD culture medium comprised 20.0 g glucose, 10.0 g yeast powder, 20.0 g peptone, 15.0 g agar, and 1.0 L distilled water. This medium was adjusted to pH 6.0 and sterilized for 15 min at 115 °C.

Detection kits of serum total cholesterol (TC) (CV ≤ 3%), total triglycerides (TG) (CV ≤ 5%), high-density lipoprotein (HDL-c) (CV ≤ 3%), low-density lipoprotein (LDL-c) (CV ≤ 8%), total protein (TP) (CV = 1.03%), albumin (ALB) (CV = 2.3%), urea nitrogen (BUN) (CV ≤ 5%), glucose (GLU) (CV ≤ 5%), total antioxidant capacity (T-AOC) (CV ≤ 3.6%), glutathione peroxidase (GSH-Px) (CV ≤ 3.56%), superoxide dismutase (SOD) (CV ≤ 1.7%), catalase (CAT) (CV ≤ 1.7%), and malondialdehyde (MDA) (CV = 1.5%) were purchased from Nanjing Jiancheng Bioengineering Institute (Nanjing, China).

### 2.2. The Preparation of Fermented Basal Diet

The *Lactobacillus amylovorus* TZR-PI001, *Enterococcus faecalis* TZR-PI003, *Bacillus subtilis* ATCC 19659, and *Candida utilis* CICC 1314 strains were taken out from the −80 °C refrigerator and activated on solid culture medium. After purification twice, they were respectively inoculated into a liquid culture medium and adjusted to a concentration of 10^8^ CFU/mL to prepare the seed culture solution. *L. amylovorus* and *E. faecalis* used MRS culture medium, *B. subtilis* used TSA culture medium, and *C. utilis* used YPD culture medium. *L. amylovorus*, *E. faecalis*, *B. subtilis*, and *C. utilis* were mixed in a ratio of 2:1:1:1 to prepare a mixed bacterial liquid. When making fermented feed, 1 kg of complete feed was added to 21 mL of mixed bacteria liquid and 540 mL of water, and then the uniformly mixed fermented feed was put into a fermentation bag, sealed, and fermented at 30 °C for 3.5 days. The fermented feed was stirred evenly before feeding.

### 2.3. Feeding Trial

#### 2.3.1. Experimental Design, Animals, and Diets

The experiment was established according to a 2 (factors) × 2 (levels) design with multi-bacteria fermentation (fed a non-fermented diet or multi-bacteria fermentation) or CF level (fed basal diet containing 2.52% CF or 7.00% CF). A total of 36 finishing pigs (Landrace × Large White × Duroc) with a similar initial body weight (70.80 ± 5.75 kg) were randomly divided into 4 treatments of 9 barrows per group: (1) pigs fed a basal diet containing 7.00% CF (HF), (2) pigs fed a basal multi-bacteria fermentation diet containing 7.00% CF (HFM), (3) pigs fed a basal diet containing 2.52% CF (LF), and (4) piglets fed a basal multi-bacteria fermentation diet containing 2.52% CF (LFM). The ingredients and the compositions of the diet were formulated according to NRC (2012) recommendations (Table 1).

#### 2.3.2. Feeding Management and Sample Collection

The experimental period lasted 28 days. Each pig was kept in 300 m^2^ rooms with individual pens (1.5 m length × 0.5 m width × 0.8 m height) and temperature-controlled (30 ± 1.2 °C), and fed at 08:00 and 18:00. Food and water were provided ad libitum.

Feed intake of pigs was recorded daily. The pigs without intake feed were weighed on day 0 and day 28. The pigs were measured for the backfat thickness, body length, body width, and chest circumference on day 28. On the morning of day 29, a total of 6 pigs without intake feed were selected from each group and a 10 mL blood sample was collected. The blood sample was kept for 60 min and centrifuged at 3500 × *g* for 10 min at 4 °C to harvest the serum. Serum was stored at −80 °C for biochemical analysis. Five days before the end of the test, six pigs were randomly selected collected rectal feces sample using sterile rectal swabs for three days, and stored at −80 °C for 16S rRNA gene sequencing. The License of Experimental Animals (SYXK 2014-0002) of the Animal Experimentation Ethics Committee of Southwest University (Chongqing, China) approved all experimental procedures.

#### 2.3.3. Biochemical Analysis

The presence of TC, TG, HDL-c, LDL-c, TP, ALB, BUN, GLU, T-AOC, GSH-Px, SOD, CAT, and MDA in serum was determined using colorimetric methods with a reagent kit according to the manufacturer’s instructions (Nanjing Jiancheng Institute of Bioengineering, Nanjing, Jiangsu, China).

#### 2.3.4. The 16S rRNA Gene Sequencing Analysis

According to the introduction of the Power Fecal DNA Isolation Kit (Mobio, USA), the DNA samples were extracted from samples of feces, stained using the Quant-iT Pico Green dsDNA Kit (Invitrogen Ltd., Paisley, UK), and quantified using a Nanodrop spectrophotometer (Nyxor Biotech, Paris, France). This experiment used the Illumina platform to perform paired-end sequencing of community DNA fragments, and the Greengenes database was selected. According to the QIIME2 DADA2 analysis process and Vsearch software analysis process, we performed sequence denoising or OTU clustering and output the ASV/OTU table, and finally removed the singleton ASVs/OTUs in the table. Through statistics on the ASV/OTU table, the specific composition information at each classification level of the microbial community in each sample was obtained. On this basis, different leveling depths were selected to calculate the alpha diversity index of the microbial community, including Chao1 and the observed species index representing richness, Shannon and Simpson index representing diversity, Faith’s PD index representing evolution-based diversity, Pielou’s evenness index representing evenness, and Good’s coverage index representing coverage. At the ASV/OTU level, we calculated the distance matrix of each sample and used a variety of unsupervised sorting and clustering methods, combined with corresponding statistical testing methods, to measure the difference in beta diversity between different groups and the significance of the difference. We also measured the differences in species’ abundance composition between different groups and searched for symbolic species at the species taxonomic composition level. Then, based on the 16S rRNA gene sequencing results, PICRUSt2 (Phylogenetic Investigation of Communities by Reconstruction of Unobserved States) software was used to analyze the abundance of marker gene sequences in the sample to predict the functional abundance of the sample. After obtaining the functional unit, the data were processed according to the MetaCyc Pathway Database to predict the metabolic function of the sample flora and find the differential pathways.

### 2.4. Digestion Trial

#### 2.4.1. Experimental Design, Treatment, and Sample Collection

A total of 5 test pigs were randomly selected from each group and kept in 30 m^2^ rooms, with 4 mechanically ventilated on the 5th day before the end of the trial period. One g/kg of titanium dioxide (TiO_2_) was added to their feed in advance, and their daily feces and feed were collected through the incomplete feces collection method for three days. To every 100 g feces, 10 mL of 10% H_2_SO_4_ was added, then stored at 4 °C for later use after being stirred evenly.

#### 2.4.2. Chemical Analysis

Samples of diets and feces were analyzed for the content of DM (Method 930.15; AOAC Int., 2007), CP (Method 990.03; AOAC Int., 2007), EE (ISO 6492: 1999), CF (ISO 6865: 2000), NDF (GB/T 20806-2006), ADF (NY/T 1459-2007), Ca (GB 5009.92-2016), total P(GB 5009.87-2016), and amino acids (Hitachi L-8800 automatic amino acid analyzer). All diets and feces samples were analyzed for the content of TiO_2_ using the method of Schürch et al. [20].

### 2.5. Data Calculation Formula and Statistical Analysis

The average daily weight gain, the average daily feed intake, and the ratio of feed to gain of piglets were calculated according to the following formula:Average daily gain (ADG) (kg/day) = (final weight-initial weight) (kg)/feed days (day)(1)
Average daily feed intake (ADFI) (g/day) = total feed intake (g)/feed days (day) (2)
The ratio of feed to gain (F/G) = average daily feed intake (g)/average daily gain (kg) × 0.001(3)

The apparent total tract digestibility (ATTD) of CP, CF, ADF, NDF, EE, Ca, total P, and amino acids was calculated according to the following equation:AD_nutrient_ = [1 − (nutrient_(feces)_/nutrient_(diet)_ × TiO_2(diet)_/TiO_2(feces)_] × 100%(4)
where AD_nutrient_ is the ATTD of a nutrient in the diet (%), Nutrient_(diet)_ and Nutrient_(feces)_ are the nutrient concentration in the diet and the feces samples respectively, and TiO_2(diet)_ and TiO_2(feces)_ are the TiO_2_ concentration (g/kg) in the diet and the feces samples, respectively.

The total tract digestion amount of CP, CF, EE, Ca, totalP, and amino acids was calculated according to the following formula:Total tract digestion amount (g/day) = ADFI (g/day) × Nutrient_(diet)_ × AD_nutrient_(5)

Data were analyzed according to two-way analysis of variance with CF levels (2 levels) and multi-bacteria solid-state fermentation (2 levels) using the GLM procedure (SAS Institute 222 Inc.; Cary, NC, USA). The values presented in the tables represent means and pooled standard error of the means (SEMs). The Student-Neuman-Keuls (SNK) test was performed to identify differences among groups. Significance was set at *p* < 0.05.

## 3. Results

### 3.1. Growth Performance

The effects of multi-bacteria solid-state fermentation and dietary CF level on the growth performance of pigs are shown in Table 2. Compared with the non-fermentation group, multi-bacteria solid-state fermentation increased the backfat thickness (*p* < 0.05). Compared with the low-fiber diet group, high-fiber diets increased the ADFI and decreased the bust circumference (*p* < 0.05), and tended to reduce the body length of pigs (*p* = 0.06). There was no interaction effect on ADG, ADFI, feed/gain, body length, body width, bust circumference, and backfat thickness between multi-bacteria solid-state fermentation and dietary CF levels (*p* > 0.05).

### 3.2. Nutrient and Amino Acids’ Digestibility

As shown in Table 3, compared with the low-fiber diet group, high-fiber diets decreased the ATTD of CP, CF, NDF, ADF, EE, Ca, 18 amino acids (Trp, Thr, Val, Met, Ile, Leu, Phe, Lys, His, Arg Asp, Ser, Glu, Gly, Ala, Cys, Tyr, and Pro), EAA, NEAA, and TAA, significantly (*p* < 0.05). Compared with the non-fermentation group, fermentation had a significant effect on increasing the ATTD of CP, CF, NDF, and ADF, 8 amino acids (Trp, Asp, Gly, Cys, Val, Met, Ile, and Leu), EAA, NEAA, and TAA (*p* < 0.05). No significant effect of the interaction between fermentation and CF levels on the ATTD of nutrients and amino acids was observed (*p* > 0.05).

As shown in Table 4, compared with the low-fiber diet group, high-fiber diets decreased the digestion amount of CP, EE, 14 amino acids (Trp, Thr, Ser, Glu, Gly, Ala, Val, Ile, Leu, Phe, Lys, His, Arg, and Pro), EAA, NEAA, and TAA (*p* < 0.05), but increased the digestion amount of CF, NDF, and ADF (*p* < 0.05) due to the additional crude fiber. Compared with the non-fermentation group, fermentation reduced the digestion amount of CF, NDF, and ADF (*p* < 0.05), which may be due to the degradation of fiber by fermentation, and increased the digestion amount of 5 amino acids (Trp, Glu, Gly, Val, Ile, and Leu), EAA, NEAA, and TAA (*p* < 0.05). The interaction between fermentation and CF levels had a significant influence on the digestion amount of CF (*p* < 0.05), which was reflected in the fact that the effect of fermentation on the digestion amount of CF was more obvious in high-fiber diet groups. No significant effect of the interaction between fermentation and CF levels on the digestion amounts of amino acids was observed (*p* > 0.05).

### 3.3. Serum Biochemical Parameters

As shown in Table 5, compared with the non-fermentation group, multi-bacteria solid-state fermentation decreased serum concentrations of TC, TG, LDL-c, BUN, and MDA (*p* < 0.05), and increased serum concentrations of HDL-c, TP, ABL, and GLU, and serum enzyme activities of T-AOC, GSH-Px, SOD, and CAT (*p* < 0.05). Compared with the non-fermentation group, high-fiber diets increased serum concentrations of TC, TG, LDL-c, BUN, and MDA (*p* < 0.05), and decreased serum concentrations of HDL-c, TP, ABL, and GLU, and serum enzyme activities of T-AOC, GSH-Px, SOD, and CAT (*p* < 0.05). There was no significant effect of the interaction between multi-bacteria fermentation and dietary CF level (*p* > 0.05).

### 3.4. The Rectal Microbial Composition and Function Prediction

There were 1564 ASVs/OTUs shared by the four groups (Figure 1). The HFM group had the most unique ASVs/OTUs—6542. The unique ASVs/OTUs of the LF and HFM groups were both more than that of LF and LFM groups.

Although the difference among alpha diversity indexes was not significant (*p* > 0.05), the Chao1 index, observed species index, Pielou’s index, and Shannon index of the HFM group were all higher than those of the other three groups (Figure 2).

The relative abundance of rectal microflora is shown in Table 6. At the phylum level, multi-bacteria solid-state fermentation reduced the relative abundance of *Bacteroidetes* (*p* < 0.05), and high-fiber diets increased the relative abundance of *Verrucomicrobia* (*p* < 0.05).

At the family level, multi-bacteria solid-state fermentation increased the relative abundance of *Lactobacillaceae* (*p* < 0.05) and decreased the relative abundance of *Streptococcaceae* (*p* < 0.05), and tended to increase the relative abundance of *Peptostreptococcaceae* (*p* = 0.055) and reduce the relative abundance of *S24-7* (*p* = 0.063). High-fiber diets increased the relative abundance of *Erysipelotrichaceae* (*p* < 0.05), reduced the relative abundance of *Clostridiaceae* (*p* < 0.05) and *Christensenellaceae* (*p* < 0.05) in fecal microorganisms, and tended to decrease the relative abundance of *Bifidobacteriaceae* (*p* = 0.057).

At the genus level, multi-bacteria solid-state fermentation increased the relative abundance of *Lactobacillus*, *Oscillospira*, and *Coprococcus* (*p* < 0.05) and reduced the relative abundance of *Streptococcus* (*p* < 0.05). *Lactobacillus* was the most abundant genera in HFM and LFM groups and *Streptococcus* was the most abundant genera in the HF and LF groups. High-fiber diets increased the relative abundance of *Clostridiaceae_Clostridium* and *Coprococcus* (*p* < 0.05), and decreased the relative abundance of *Akkermansia* and *Oscillospira* (*p* < 0.05). There were significant interaction effects on the relative abundance of *Oscillospira* and *Coprococcus* between multi-bacteria fermentation and dietary CF levels (*p* < 0.05).

As shown in Table 7, multi-bacteria fermentation increased the abundance of YAMINSYN3-PWY, PWY-7013, GOLPDLCAT-PWY, ARGORNPROST-PWY, and PWY-5022 pathways (*p* < 0.05), and decreased the abundance of PWY-6470, PWY0-862, HSERMETANA-PWY, LACTOSECAT-PWY, MET-SAM-PWY, PWY-6700, PWY-5347, PWY0-1061, and LACTOSECAT-PWY pathways (*p* < 0.05). High-fiber diets increased the abundance of TCA-GLYOX-BYPASS, GLYCOLYSIS-TCA-GLYOX-BYPASS, and PWY-6906 pathways (*p* < 0.05). There was no significant effect of the interaction on the associated nutrients’ metabolic pathways of rectal microorganisms between multi-bacteria fermentation and dietary CF levels (*p* > 0.05).

## 4. Discussion

Multi-bacteria fermentation and dietary CF levels had little effect on the weight gain of finishing pigs in this study. The effects of crude fiber and fermented feed on growth performance varied in different experiments. Feeding *Bacillus* fully fermented feed significantly increased the ADG of weaned piglets and reduced the F:G [21], while adding fermented grain feed or flaxseed meal to piglets’ diet did not significantly affect growth performance and ATTD in other trials [22,23].

Dietary CF levels showed the most important impact in this study. Compared with the 2.52% CF diet, the 7.00% diet significantly reduced the ATTD and digestion amount of CP, CF, NDF, ADF, EE, and amino acids, indicating that the 7.00% CF level exceeded the tolerance of the finishing pigs in this experiment. Excess crude fiber would increase the flow of the digestive tract, and the soluble fiber with high water-holding capacity wrapped the chyme and caused insufficient contact between the contents and the intestinal wall, resulting in a reduction of nutrient digestibility [24]. This result was consistent with the results of Chen et al. [25], in whose study the CF level of 10% alfalfa (four CF levels were set at 0%, 5%, 10%, and 20%) was similar to our experiment and the total tract digestibility of CP, DM, ADF, and NDF in the 10% alfalfa diet group was significantly lower than that in the 0% and 5% group. Asosta et al. [26] applied distillers dried grains with soluble (DDGS) in a gradient, and the apparent ileal digestibility (AID) of each amino acid was reduced to varying degrees, similar to the results of this study. The optimum fiber tolerance of finishing pigs in our experiment was less than 7%, and the follow-up experiment continued to explore this. Solid-state fermentation was able to improve the nutrient digestion of CP, CF, NDF, ADF, EAA, NEAA, and TAA, and alleviate the side effect of high fiber on the nutrients. *Enterococcus faecalis* is a common lactic acid bacteria. Solid-state fermentation with *Enterococcus faecalis* was able to reduce pH, destroy plant cell walls, and increase the content of soluble dietary fiber [27]. *Bacillus subtilis* were capable of secreting xylanase and *β*-mannanase, decomposing plant hemicellulose, and reducing the content of non-starch polysaccharides (NSP) and anti-nutritional factors in grain feeds [28,29]. *Candida utilis* could decompose lignin and cellulose into monosaccharides and ferment them under aerobic conditions as well. *Candida utilis* was particularly efficient in metabolizing hexoses [30]. Solid-state fermentation could promote nutrient digestion and utilization of crude fiber.

Usually, high-fiber diets can reduce blood sugar, blood lipids, and cholesterol [31,32], and the fermentability of crude fiber leads to an increase in the production of colonic short-chain fatty acids (SCFA), including the increase of the ratio of propionate and butyrate and the reduction of the ratio of acetate [33]. Acetate participates in de novo lipogenesis through the conversion to acetyl-CoA, while propionate participates in gluconeogenesis and was regarded as an inhibitor of fat formation. Butyrate plays an important role in maintaining the integrity of the mucosa and controlling inflammation [34]. Besides, SCFAs affected the activity of metabolic hormones such as leptin, insulin, and glucagon [35]. However, the serum TC, TG, and LDL of the high-fiber diet group in this experiment were significantly higher than those of the low-fiber diet group, which contradicted the above conclusions and might be due to a metabolic disorder caused by a too-high CF level. Previous studies found that there was a strong negative correlation between branched-chain amino acids and TC [36], and the ATTD of leucine, isoleucine, and valine in the high-fiber diet group was significantly lower than that in the low-fiber diet group in this experiment, which was consistent with this finding. High-fiber diets increased the content of serum BUN, and decreased the ATTD of glutamic acid, arginine, proline, and aspartic acid, which are involved in the urea cycle. These results indicated that a high-fiber diet of up to 7.00% could affect the urea cycle and nitrogen balance in finishing pigs and reduce the utilization of nitrogen. However, solid-state fermentation was able to alleviate the adverse effects of high-fiber diets in many ways, such as reducing the content of serum TC and TG, increasing antioxidant capacity, and improving the utilization of nitrogen.

In general, high-fiber diets and fermented feed had a positive effect on the composition and diversity of pig fecal microorganisms and the relative abundance of beneficial bacteria. The HF group had the largest number of species, and the highest species richness and uniformity. Through microbial community analysis, *Firmicutes*, accounting for 70–80% of the total, was still the most important phyla in the four groups at the phylum level, followed by *Bacteroidetes*. The relative abundance of *Firmicutes* in the fermented feed group was relatively high, which was related to the fact that *Lactobacillus amylovorus*, *Enterococcus faecalis*, and *Bacillus subtilis* participating in fermentation all belong to *Firmicutes*. At the same time, *Firmicutes* were the main degradation bacteria of macromolecules such as cellulose and resistant starch. The produced SCFAs, especially butyric acid, mainly provide energy for the host colonic epithelial cells, and also have a probiotic effect, such as anti-inflammatory and antioxidant [37]. Studies have shown that the body fat rate of pigs is related to the relative abundance of *Firmicutes* and *Bacteroidetes*. *Bacteroidetes* are common microorganisms that degrade polysaccharides [38]. Researchers found that removing *Bacteroidetes* could promote the secretion of glucagon-like peptide 1 [39], and the relative abundance of *Bacteroidetes* in obese pigs was lower than that in lean pigs [40]. In this experiment, feeding fermented feed significantly reduced the relative abundance of *Bacteroidetes*, and the backfat thickness of fermentation groups was significantly higher than that of the unfermented feed group, which is in accordance with the above conclusion and is also similar to the experimental results of Cui et al. [41]. The experiment of Cui et al. also proved that the effect of probiotics on lipid metabolism enzyme mRNAs is the opposite in subcutaneous fat and liver, which explains why fermented feed reduced serum TC and TG while increasing the backfat depth of finishing pigs in this experiment.

At the family level, *Lactobacillaceae* and *Ruminococcaceaes* were the two dominant groups, and the multi-bacteria fermented feed significantly improved the relative abundance of *Lactobacillaceae*. Previous studies revealed that *Lactobacillaceae* was involved in the biotransformation of secondary bile acids [42,43,44]. Compared with primary bile acids, secondary bile acids are less soluble and easier to be excreted. Large excretion of secondary bile acids required the supplement of cholesterol from de novo synthesis. Therefore, *Lactobacillaceae* had the potential of reducing cholesterol [45], which was consistent with the results of serum biochemical indicators in this experiment. *Lactobacillaceae* was able to secrete lactic acid and decrease the pH in the intestines, which is beneficial to the degradation of macromolecular substances. Multi-bacteria fermentation tended to increase the relative abundance of *Peptostreptococcaceae* and reduce the relative abundance of *S24-7*. *Peptostreptococcaceae,* as an intestinal symbiotic bacteria, has been proven to be able to produce volatile fatty acids, and maintain the host intestinal homeostasis and energy metabolism [46]. *S24-7* is a type of microorganism related to intra-abdominal infections [47].

At the genus level, *Lactobacillus* and *Streptococcus* were the two dominant groups in the fermented feed group and the non-fermented feed group respectively, similar to the family level. Fermentation significantly reduced the abundance of the pathogenic bacteria *Streptococcus*. The abundance of *Clostridiaceae_Clostridium* was closely related to dietary protein levels and is usually the most important genus in high-protein diets [48]. High-fiber diets significantly reduced the relative abundance of *Clostridiaceae_Clostridium*, which was presumably due to the fact that excessive fiber concentration leads to excessive protein loss in the digestive tract and a decrease in fermentable protein substrates [49]. Although the difference in the abundance of *Bifidobacterium* at the genus level was not significant, *Bifidobacterium* was still a dominant species in the LF group. *Bifidobacterium* had an active role in anti-inflammatory and anti-pathogenic microorganisms and prevented pathogenic infection by producing acetate [50]. As the dominant species in the HF group, *Coprococcus* was a kind of beneficial bacteria that can use carbohydrates to produce butyrate [49] to achieve probiotic effects. Both fermentation and high-fiber diet had a significant effect on it, and the interaction was significant as well. *Oscillospira* is a biomarker of the LN group and a kind of butyric acid-producing bacteria as well, but the substrate glucuronate of *Oscillospira* is a common host-derived sugar [51], which makes *Oscillospira* related to body thinness [52], and studies have shown that the relative abundance of *Oscillospira* in the cecum of birds, fish, and mice increased after long-term fasting [53].

In terms of function prediction, feeding fermented feed had a significant effect on metabolic pathways associated with synthesis, degradation, and transformation of substances in finishing pigs. For example, fermented feed promoted the biosynthesis of polyamine (YAMINSYN3-PWY), which affects RNA and DNA structure, ribosome function, and the activities of many enzymes, including kinase and phosphatase, and plays a role in controlling membrane potential and potassium homeostasis in glutamate receptor ion channels [54]. Fermented feed also increased the pathways of ARGORNPROST-PWY (arginine, ornithine, and proline interconversion), GOLPDLCAT-PWY (glycerol degradation), PWY-7013 (propanediol degradation), and PWY-5022 (4-aminobutanoate degradation), which meant improvements of urea cycle efficiency and fatty acid utilization. In addition, the fermented feed was able to inhibit the biosynthesis of peptidoglycan (PWY-6470) and queuosine (PWY-6700). Peptidoglycan was the main component of the cell wall of many cocci [55], and queuosine was the raw material of DNA and RNA, indicating that fermented feed has the potential to inhibit pathogens. High-fiber diets were able to increase the pathway of glycolysis, pyruvate dehydrogenase, TCA, and glyoxylate bypass (TCA-GLYOX-BYPASS and GLYCOLYSIS-TCA-GLYOX-BYPASS) to improve the catabolism of soluble sugar and promote chitin derivatives’ degradation (PWY-6906), which has an inhibition effect on pathogenic bacteria such as *Escherichia coli* and *Salmonella typhimurium* [56].

## 5. Conclusions

Multi-bacteria solid-state fermentation and dietary CF levels had little effect on the weight gain of finishing pigs. Dietary 7.00% CF had a negative effect on ATTD and digestion amount, but multi-bacteria solid-state fermentation diets could relieve this negative effect and increase backfat depth. High-fiber diets and multi-bacteria solid-state fermentation improved the diversity and abundance of fecal microorganisms in finishing pigs. Feeding a high-fiber, multi-bacteria, solid-state fermented diet increased the relative abundance of the beneficial bacteria *Coprococcus* and *Lactobacillus*, and reduced the relative abundance of *Streptococcus* and *Oscillospira*. Multi-bacteria solid-state fermentation helped to increase the abundance of metabolic pathways associated with nutrients’ utilization in the fecal microorganism.

## Figures and Tables

**Figure 1 animals-11-03079-f001:**
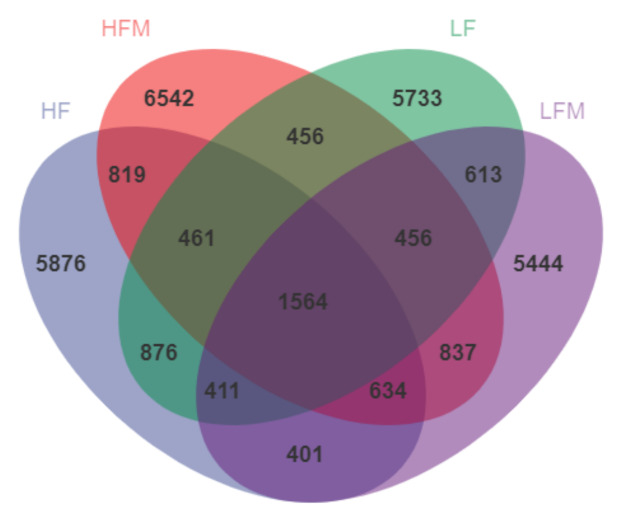
ASV/OTU Venn diagram.

**Figure 2 animals-11-03079-f002:**
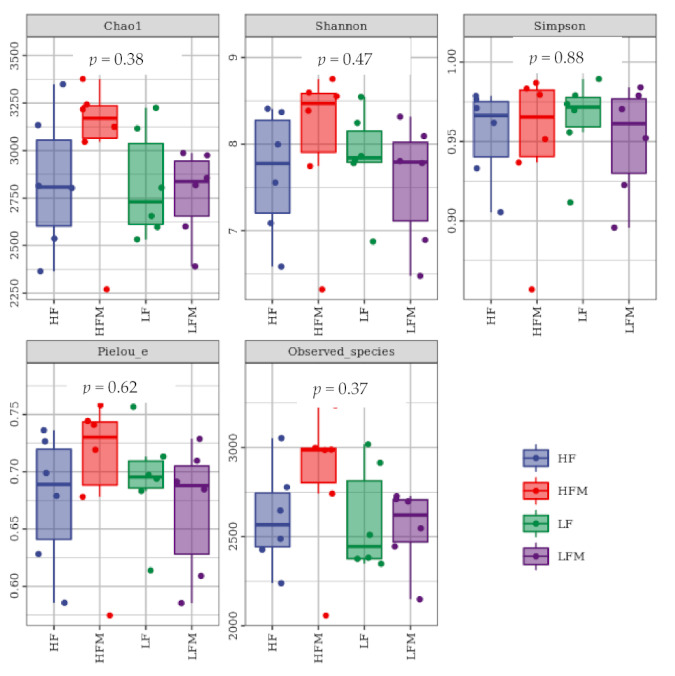
Alpha index box plot.

**Table 1 animals-11-03079-t001:** Composition of experimental diets (based on dry matter, %).

Ingredients	2.52% CF	7.00% CF	Nutritional Composition	2.52% CF	7.00% CF
Corn (CP 8.7%, DM > 86%)	74.91	53.36	DE (MJ/kg)	13.71	13.71
Soybean meal (CP 47.8%)	15.65	14.67	CP (%)	15.00	15.00
Wheat bran	5.63	1.00	CF (%)	2.52	7.00
Rice bran	1.00	5.68	ADF (%)	3.72	8.13
Alfalfa meal (CP 14%, CF 30%)	0.00	17.57	NDF (%)	10.66	14.78
Fat powder	0.00	5.70	Ca (%)	0.52	0.52
Monocalcium phosphate	0.59	0.56	Total P (%)	0.47	0.47
Limestone	0.86	0.16	Avail. P (%)	0.24	0.24
Salt	0.30	0.30	Trp (%)	0.17	0.17
Trace mineral and vitamin premix ^a^	1.00	1.00	Lys (%)	0.73	0.73
L-Lysine·HCl	0.06	0.00	Thr (%)	0.54	0.54
Total	100.0	100.0	Met + Cys (%)	0.50	0.50

Note: ^a^ Provided the following per kilogram of diet: 150 IU of vitamin D, 1300 IU of vitamin A, 11 IU of vitamin E, 7 mg of Pantothenic, 0.5 mg of vitamin K, 2 mg of Riboflavin, 0.3 mg of Folic Acid, 30 mg of Niacin, 0.05 mg Biotin, 1.0 mg of Thiamine, 1.0 mg of vitamin B6, 5 mg of vitamin B12, and 0.3 g Choline. Provided the following per kilogram of diet: 40 mg Fe, 50 mg Zn, 3 mg Cu, 2 mg Mn, 0.15 mg Se, 0.14 mg.

**Table 2 animals-11-03079-t002:** Effects of multi-bacteria fermentation and dietary CF level on growth performance in finishing pigs.

	Treatments ^1^				SEM	*p*-Value		
Item	HF	HFM	LF	LFM		Fermentation	CF Levels	Fermentation × CF Levels
Initial weight (kg)	70.8	70.9	70.8	70.8	1.44	0.968	0.994	0.989
Final weight (kg)	99.4	100.2	100.0	96.1	1.82	0.555	0.505	0.367
ADG (kg/d)	1.02	1.02	1.03	1.01	0.02	0.940	0.764	0.600
ADFI (g)	3463	3464	3421	3427	18	0.868	0.039	0.882
Feed/Gain (g/g)	3.40	3.40	3.33	3.39	0.08	0.734	0.646	0.704
Body length (cm)	107.6	106.3	111.6	108.9	1.7	0.256	0.060	0.671
Body width (cm)	63.2	62.9	62.0	60.3	1.2	0.427	0.138	0.595
Chest circumference (cm)	105.2	106.1	107.7	110.0	1.4	0.251	0.028	0.604
Backfat thickness (cm)	9.22	11.56	9.00	11.44	0.66	0.001	0.803	0.934

^1^ Pigs fed a basal diet containing 7.00% CF (HF), pigs fed a basal multi-bacteria fermentation diet containing 7.00% CF (HFM), pigs fed a basal diet containing 2.52% CF (LF), and piglets fed a basal multi-bacteria fermentation diet containing 2.52% CF (LFM). The names of the 4 treatments in the following tables and figures are the same as those in this table.

**Table 3 animals-11-03079-t003:** Effects of multi-bacteria fermentation and dietary CF level on apparent total tract digestibility (ATTD) of CP, CF, EE, Ca, Total *p*, and amino acids in finishing pigs (%).

	Treatments				SEM	*p*-Value		
Items	HF	HFM	LF	LFM		Fermentation	CF Levels	Fermentation × CF Levels
Nutrient substance								
CP	78.24	81.30	84.56	85.64	0.71	0.020	<0.001	0.205
CF	44.31	48.76	50.02	52.11	1.29	0.035	0.008	0.389
NDF	48.81	51.81	55.41	57.30	0.91	0.028	<0.001	0.558
ADF	43.50	47.18	49.88	52.76	1.34	0.040	0.002	0.772
EE	71.71	74.06	78.21	78.17	0.71	0.143	<0.001	0.129
Ca	73.26	79.04	83.50	83.68	2.78	0.316	0.028	0.343
Total *p*	53.70	56.08	54.86	55.27	2.84	0.634	0.951	0.736
Amino acids								
Trp	73.48	77.84	84.32	86.15	0.84	0.006	<0.001	0.083
Asp	74.89	78.87	81.40	83.33	0.45	<0.001	<0.001	0.054
Thr	79.52	82.41	85.23	86.53	1.17	0.112	0.003	0.516
Ser	74.99	77.73	82.72	84.29	1.09	0.084	0.001	0.601
Glu	81.65	84.21	88.36	89.15	0.88	0.093	<0.001	0.343
Gly	77.82	81.15	85.58	87.41	0.60	0.003	<0.001	0.247
Ala	81.63	83.77	86.19	87.12	1.12	0.208	0.008	0.606
Cys	81.31	83.97	87.63	89.52	0.40	<0.001	<0.001	0.366
Val	83.08	85.68	88.48	89.72	0.53	0.007	<0.001	0.233
Met	81.59	83.52	87.02	88.46	0.33	0.001	<0.001	0.481
Ile	80.38	82.95	85.59	86.99	0.48	0.003	<0.001	0.259
Leu	86.27	88.19	89.89	91.19	0.64	0.037	0.001	0.646
Tyr	76.87	79.82	82.42	83.21	1.28	0.181	0.008	0.423
Phe	73.73	77.12	81.18	81.27	1.11	0.151	0.001	0.178
Lys	78.27	80.28	85.76	87.41	0.92	0.082	<0.001	0.849
His	86.82	87.85	90.60	92.41	0.65	0.061	0.001	0.568
Arg	88.83	90.22	92.58	93.48	0.61	0.099	0.001	0.702
Pro	79.20	83.28	89.53	90.70	2.48	0.320	0.007	0.573
EAA	82.52	84.77	87.78	89.27	0.65	0.021	<0.001	0.579
NEAA	79.06	82.12	86.19	87.68	0.68	0.045	<0.001	0.437
TAA	80.94	83.55	87.04	88.53	0.79	0.032	<0.001	0.497

EAA represents essential amino acids, including Trp, Thr, Val, Met, Ile, Leu, Phe, Lys, His, and Arg; NEAA represents non-essential amino acids, including Asp, Ser, Glu, Gly, Ala, Cys, Tyr, and Pro; TAA represents total amino acids.

**Table 4 animals-11-03079-t004:** Effects of multi-bacteria fermentation and dietary CF level on total tract digestion amount of CP, CF, EE, Ca, Total *p*, and amino acids in finishing pigs (g/day).

	Treatments				SEM	*p*-Value		
Item	HF	HFM	LF	LFM		Fermentation	CF Levels	Fermentation × CF Levels
Nutrient substance								
CP	411.55	412.01	437.06	429.77	3.68	0.381	<0.001	0.323
CF	123.39	100.99	48.26	38.84	2.75	<0.001	<0.001	0.046
NDF	249.85	188.95	202.10	146.48	3.98	<0.001	<0.001	0.525
ADF	124.13	103.77	63.32	48.46	2.75	<0.001	<0.001	0.346
EE	264.80	273.50	284.86	285.70	2.61	0.105	<0.001	0.171
Ca	16.22	17.11	16.86	17.59	0.45	0.112	0.254	0.873
Total *p*	7.49	7.48	7.69	7.30	0.39	0.619	0.976	0.633
Amino acids								
Trp	4.35	4.62	4.93	5.07	0.05	0.004	<0.001	0.209
Asp	20.25	21.79	20.19	21.02	0.12	0.091	0.110	0.121
Thr	14.90	15.42	15.79	16.03	0.22	0.120	0.009	0.545
Ser	9.19	9.29	10.27	10.25	0.13	0.769	<0.001	0.673
Glu	37.48	39.26	44.98	45.20	0.41	0.041	<0.001	0.094
Gly	14.55	15.22	16.11	16.57	0.11	0.001	<0.001	0.384
Ala	22.81	23.00	23.65	24.14	0.31	0.304	0.013	0.634
Cys	5.69	6.20	6.11	6.47	0.03	0.305	0.247	0.333
Val	20.98	21.84	22.18	22.57	0.13	0.002	<0.001	0.111
Met	8.47	8.72	8.93	9.34	0.03	0.486	0.565	0.339
Ile	16.26	16.66	17.01	17.45	0.10	0.003	<0.001	0.856
Leu	42.43	43.43	43.70	44.47	0.32	0.023	0.006	0.716
Tyr	8.46	8.81	8.93	8.92	0.14	0.253	0.074	0.241
Phe	11.41	11.97	12.35	12.51	0.17	0.072	0.003	0.286
Lys	19.66	19.43	21.45	21.52	0.23	0.744	<0.001	0.519
His	11.17	11.20	11.48	11.91	0.08	0.322	<0.001	0.341
Arg	21.46	21.72	21.97	22.12	0.15	0.203	0.014	0.704
Pro	20.06	20.85	25.20	25.14	0.64	0.585	<0.001	0.526
EAA	171.07	175.00	179.79	183.46	1.35	0.023	<0.001	0.927
NEAA	138.50	144.41	155.43	158.19	1.69	0.033	<0.001	0.378
TAA	309.57	319.41	335.22	341.65	3.02	0.027	<0.001	0.588

EAA represents essential amino acids, including Trp, Thr, Val, Met, Ile, Leu, Phe, Lys, His, and Arg; NEAA represents non-essential amino acids, including Asp, Ser, Glu, Gly, Ala, Cys, Tyr, and Pro; TAA represents total amino acids.

**Table 5 animals-11-03079-t005:** Effects of multi-bacteria fermentation and dietary CF level on serum biochemical parameters in finishing pigs.

	Treatments				SEM	*p*-Value		
Items	HF	HFM	LF	LFM		Fermentation	CF Levels	Fermentation × CF Levels
TC (mmol/L)	5.113	4.478	3.919	3.543	0.201	0.014	<0.001	0.514
TG (mmol/L)	2.703	2.163	1.991	1.844	0.064	<0.01	<0.001	0.204
LDL-c (mmol/L)	2.51	2.08	1.67	1.33	0.17	0.044	0.001	0.813
HDL-c (mmol/L)	0.60	0.74	1.07	1.18	0.05	0.037	<0.001	0.726
TP (gprot/L)	34.88	39.52	43.77	45.17	1.53	0.041	<0.001	0.287
ABL (g/L)	19.24	26.11	28.04	33.45	1.20	<0.001	<0.001	0.536
BUN (mmol/L)	11.78	9.65	8.69	8.42	0.32	<0.001	<0.001	0.355
GLU (mmol/L)	3.82	4.29	5.01	5.50	1.02	<0.001	<0.001	0.916
T-AOC (mmol/gprot)	0.33	0.46	0.52	0.54	0.03	0.042	0.002	0.145
MDA (nmol/mL)	7.88	6.79	5.14	4.82	0.32	0.049	<0.001	0.253
GSH-Px (U/mgprot)	145.86	177.78	205.76	207.09	5.61	0.012	<0.001	0.318
SOD (U/mL)	40.26	48.99	68.16	82.27	3.00	0.003	<0.001	0.388
CAT (U/mL)	5.63	8.45	9.21	9.49	0.68	0.041	0.005	0.286

ETC: total cholesterol; TG: triglyceride; LDL-c: low-density lipoprotein-cholesterol; HDL-c: high-density lipoprotein-cholesterol; TP: total protein; ABL: albumin; GLU: glucose; T-AOC: total antioxidant capacity; MDA: malondialdehyde; GSH-Px: glutathione peroxidase; SOD: superoxide dismutase; CAT: catalase.

**Table 6 animals-11-03079-t006:** Effects of multi-bacteria fermentation and dietary CF level on the relative abundance of rectal microorganisms (%).

Items	Treatments				SEM	*p*-Value		
	HF	HFM	LF	LFM		Fermentation	CF Levels	Fermentation × CF Levels
Phylum								
*Firmicutes*	75.34	79.77	71.52	79.33	4.97	0.232	0.672	0.737
*Bacteroidetes*	19.63	9.31	19.53	7.27	4.48	0.020	0.814	0.831
*Spirochaetes*	1.96	7.36	6.16	9.04	3.00	0.183	0.339	0.679
*Actinobacteria*	1.10	0.69	1.04	2.77	0.56	0.246	0.083	0.069
*Verrucomicrobia*	0.55	1.08	0.20	0.44	0.22	0.102	0.039	0.539
Family								
*Lactobacillaceae*	18.50	32.29	5.48	25.68	7.41	0.033	0.200	0.670
*Ruminococcaceae*	14.32	15.18	20.24	16.02	2.03	0.418	0.112	0.226
*Clostridiaceae*	6.22	7.43	8.80	14.54	2.29	0.145	0.047	0.335
*Lachnospiraceae*	8.90	11.86	7.95	8.23	2.28	0.487	0.327	0.564
*S24-7*	13.26	5.47	12.14	2.25	4.49	0.063	0.635	0.817
*Streptococcaceae*	16.54	0.02	15.10	0.02	4.41	0.002	0.872	0.872
*Spirochaetaceae*	1.95	7.34	6.15	9.03	3.00	0.183	0.338	0.680
*Christensenellaceae*	0.77	0.54	2.65	2.57	0.63	0.804	0.005	0.906
*Prevotellaceae*	1.32	0.99	1.77	0.59	0.40	0.077	0.952	0.300
*Bifidobacteriaceae*	0.73	0.19	0.82	2.34	0.56	0.394	0.057	0.078
*[Mogibacteriaceae]*	0.79	0.99	1.09	0.89	0.18	0.999	0.569	0.269
*Peptostreptococcaceae*	0.55	0.65	0.58	1.30	0.23	0.055	0.105	0.147
*Erysipelotrichaceae*	1.01	1.05	0.38	0.50	0.21	0.660	0.004	0.848
Genus								
*Lactobacillus*	18.43	31.49	5.47	25.01	7.35	0.038	0.201	0.664
*Streptococcus*	15.23	0.01	14.69	0.02	4.22	0.002	0.950	0.949
*Treponema*	1.95	7.34	6.15	9.03	3.00	0.183	0.338	0.680
*SMB53*	3.32	3.38	3.19	6.84	1.32	0.174	0.219	0.188
*Oscillospira*	1.35	1.52	4.59	2.10	0.50	0.031	0.001	0.015
*Prevotella*	1.28	0.98	1.72	0.56	0.40	0.082	0.976	0.293
*Bifidobacterium*	0.73	0.19	0.82	2.34	0.56	0.394	0.057	0.078
*Ruminococcus*	0.71	1.30	0.81	0.80	0.23	0.220	0.394	0.204
*Clostridiaceae_clostridium*	0.48	0.51	0.97	0.91	0.21	0.939	0.049	0.825
*Coprococcus*	0.44	1.17	0.33	0.34	0.14	0.018	0.004	0.022
*Akkermansia*	0.55	1.01	0.14	0.37	0.21	0.116	0.022	0.598

**Table 7 animals-11-03079-t007:** Effects of multi-bacteria fermentation and dietary CF level on the relative abundances of the associated nutrients’ metabolic pathways of rectal microorganisms.

Items	Treatments				SEM	*p*-Value		
	HF	HFM	LF	LFM		Fermentation	CF Levels	Fermentation × CF Levels
PWY-6470	729.61	8.14	444.15	10.87	82.13	<0.001	0.101	0.095
POLYAMINSYN3-PWY	52.15	135.70	45.15	88.13	17.97	0.002	0.145	0.272
PWY0-862	605.45	66.60	611.97	29.62	142.56	0.001	0.916	0.880
PWY-7013	33.07	99.93	51.77	83.94	12.84	0.001	0.917	0.192
HSERMETANA-PWY	692.43	229.51	683.21	320.10	73.96	<0.001	0.588	0.508
LACTOSECAT-PWY	474.18	44.85	377.46	30.27	105.74	0.002	0.604	0.702
MET-SAM-PWY	930.42	318.53	866.11	464.23	96.80	<0.001	0.679	0.291
PWY-6700	869.73	419.83	863.10	484.51	75.65	<0.001	0.705	0.642
PWY-5347	1002.32	424.48	1000.06	597.52	85.30	<0.001	0.329	0.316
PWY0-1061	723.03	196.47	685.86	190.76	122.65	<0.001	0.863	0.899
GOLPDLCAT-PWY	19.01	42.82	30.62	47.65	6.36	0.004	0.211	0.600
TCA-GLYOX-BYPASS	18.25	15.49	0.98	7.17	5.43	0.755	0.029	0.419
GLYCOLYSIS-TCA-GLYOX-BYPASS	34.80	29.72	1.95	14.13	10.28	0.733	0.029	0.411
ARGORNPROST-PWY	127.16	191.47	132.46	247.88	20.47	<0.001	0.147	0.226
PWY-5022	86.83	111.44	95.56	162.32	18.54	0.023	0.124	0.269
LACTOSECAT-PWY	474.18	44.85	377.46	30.27	105.74	0.002	0.604	0.702
PWY-6906	0.35	0.26	0.03	0.00	0.09	0.514	0.006	0.739
PWY-6470	729.61	8.14	444.15	10.87	82.13	<0.001	0.101	0.095
POLYAMINSYN3-PWY	52.15	135.70	45.15	88.13	17.97	0.002	0.145	0.272
PWY0-862	605.45	66.60	611.97	29.62	142.56	0.001	0.916	0.880
PWY-7013	33.07	99.93	51.77	83.94	12.84	0.001	0.917	0.192
HSERMETANA-PWY	692.43	229.51	683.21	320.10	73.96	<0.001	0.588	0.508
LACTOSECAT-PWY	474.18	44.85	377.46	30.27	105.74	0.002	0.604	0.702
MET-SAM-PWY	930.42	318.53	866.11	464.23	96.80	<0.001	0.679	0.291
PWY-6700	869.73	419.83	863.10	484.51	75.65	<0.001	0.705	0.642
PWY-5347	1002.32	424.48	1000.06	597.52	85.30	<0.001	0.329	0.316
PWY0-1061	723.03	196.47	685.86	190.76	122.65	<0.001	0.863	0.899
GOLPDLCAT-PWY	19.01	42.82	30.62	47.65	6.36	0.004	0.211	0.600
TCA-GLYOX-BYPASS	18.25	15.49	0.98	7.17	5.43	0.755	0.029	0.419

PWY-6470 associated with peptidoglycan biosynthesis V (and beta-lactam resistance); POLYAMINSYN3-PWY associated with super pathway of polyamine biosynthesis II; PWY0-862 associated with (5Z)-dodec-5-enoate biosynthesis; PWY-7013 associated with L-1,2-propanediol degradation; HSERMETANA-PWY associated with L-methionine biosynthesis III; LACTOSECAT-PWY associated with lactose and galactose degradation I; MET-SAM-PWY associated with super pathway of S-adenosyl-L-methionine biosynthesis; PWY-6700 associated with queuosine biosynthesis; PWY-5347 associated with super pathway of L-methionine biosynthesis (trans-sulfuration); PWY0-1061 associated with super pathway of L-alanine biosynthesis; GOLPDLCAT-PWY associated with super pathway of glycerol degradation to 1,3-propanediol; TCA-GLYOX-BYPASS associated with super pathway of glyoxylate bypass and TCA; GLYCOLYSIS-TCA-GLYOX-BYPASS associated with super pathway of glycolysis, pyruvate dehydrogenase, TCA, and glyoxylate bypass; ARGORNPROST-PWY associated with arginine, ornithine, and proline interconversion; PWY-5022 associated with 4-aminobutanoate degradation V; LACTOSECAT-PWY associated with lactose and galactose degradation I; PWY-6906 associated with chitin derivatives’ degradation.

## Data Availability

The data presented in this study were available on request from first author (Hu P.) and the corresponding author (Tang Z.).

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
