# Peer review of "Effects of Multi-Bacteria Solid-State Fermented Diets with Different Crude Fiber Levels on Growth Performance, Nutrient Digestibility, and Microbial Flora of Finishing Pigs"

_animals, 2021, doi:10.3390/ani11113079_

Round 1
Reviewer 1 Report
The study entitled the effects of multi-bacteria solid-state fermented diets with different crude fiber levels on growth performance, nutrient digestibility, and microbial flora of finishing pigs was designed to investigate with the two main factors multi-bacteria solid-state fermentation (fed non-fermented diet or multi-bacteria fermentation) or crude fiber levels (fed a basal diet containing 2.52% CF or 7.00% CF) in 2 by 2 factorial arrangement of treatment. The authors did not elicit a response of multi-bacteria solid-state fermentation and dietary CF levels on weight gain of finishing pigs. The 7.00% CF had a negative effect on ATTD and digestion amount. High-fiber diets and multi-bacteria solid-state fermentation improved the diversity and abundance of fecal microorganisms in finishing pigs.
I do not recommend publication of the manuscript due to the following reasons:
What is the control of the experiment and how do you justify the control? What is the experimental unit of this study? What is the experimental design for the study? There is nothing like factorial design. Poor description of study timeline making repetition of the study very difficult.
In the introduction, the study objective is not stated and justified. No hypothesis was stated. The presentation of the facts is not coherent and cohesive. The motivation of the study is not known.
Very poor presentation of results. One example is in lines 43 to 51. What two results are being compared? Correct this and elsewhere in the entire manuscript.
Line 11: Dietary fiber is not a nutrient. Please modify the statement.
Lines 14 – 17: The treatments abbreviations are ambiguous and very difficult for readers to follow. Instead used the following abbreviations for the treatments: (1) pigs fed a basal diet containing 7.00% CF (HF) instead of (HN); (2) pigs fed a basal multi-bacteria fermentation diet containing 7.00% CF (HFM) instead of (HF); (3) pigs fed a basal diet containing 2.52% CF (LF) instead of (LN); (4) piglets fed a basal multi-bacteria fermentation diet containing 2.52% CF (LFM) instead of (LF)
Line 17: There is nothing called factorial design but there is factorial arrangement of the treatment structure.
Line 28: Do not start sentences with figure. You could start the sentence like this: A total of 36 pigs…………. Correct this in the entire manuscript.
Line 29: Pig numbers in line 29 contradicts that of line 28.
Lines 32 – 43. This sentence is too long. Break it into 3 or 4 sentences. Correct this long sentences in the entire manuscript.
Lines 53: ATTD of what?
Lines 70 – 71: Is it fermentation of microorganisms or fermentation by microorganisms?
Lines 104 – 109: what is the coefficient of variation (CV) of the biochemical analysis?
Line 132: Delete the phrase “according to” and replace with “to meet or exceed”
Line 133: The nutritional composition information in table 1, is it analyzed values or calculated values?
Lines 147 to 149: These body measurements were taken at day 28 why not on day 0 as well. The difference that were observed on day 28 could have also existed at day 0. Therefore, there will a need for covariate analysis.
Line 474: Did authors investigate any metabolic pathway? This is over generalization.
Author Response
Section: "Responses to the Comments by reviewer 1"
General Comment: The study entitled the effects of multi-bacteria solid-state fermented diets with different crude fiber levels on growth performance, nutrient digestibility, and microbial flora of finishing pigs was designed to investigate with the two main factors multi-bacteria solid-state fermentation (fed non-fermented diet or multi-bacteria fermentation) or crude fiber levels (fed a basal diet containing 2.52% CF or 7.00% CF) in 2 by 2 factorial arrangement of treatment. The authors did not elicit a response of multi-bacteria solid-state fermentation and dietary CF levels on weight gain of finishing pigs. The 7.00% CF had a negative effect on ATTD and digestion amount. High-fiber diets and multi-bacteria solid-state fermentation improved the diversity and abundance of fecal microorganisms in finishing pigs.
I do not recommend publication of the manuscript due to the following reasons:
What is the control of the experiment and how do you justify the control? What is the experimental unit of this study? What is the experimental design for the study? There is nothing like factorial design. Poor description of study timeline making repetition of the study very difficult.
In the introduction, the study objective is not stated and justified. No hypothesis was stated. The presentation of the facts is not coherent and cohesive. The motivation of the study is not known.
Very poor presentation of results. One example is in lines 43 to 51. What two results are being compared? Correct this and elsewhere in the entire manuscript.
Authors'responses and locations of the revisions: Thanks Reviewer giving more suggestions. the experimental design is 2 (factors) × 2 (levels) design with the two factors and 2 (levels): multi-bacteria solid-state fermentation (fed non-fermented diet or multi-bacteria fermentation) , CF levels (fed a basal diet containing 2.52% CF or 7.00% CF).
We have rerevised “the introduction” and “results”.“results”: this study aimed To study the effects of multi-bacteria solid-state fermented diets and dietary crude fiber levels on finishing pigs, so we showed one factor- multi-bacteria solid-state fermentation-effect, another factor dietary crude fiber levels-effect, and the interaction between multi-bacteria fermentation and dietary CF levels.
Comment 1: In the introduction, the study objective is not stated and justified. No hypothesis was stated. The presentation of the facts is not coherent and cohesive. The motivation of the study is not known.
Authors'responses and locations of the revisions: Thanks. Owing to your suggestion, we added relevant arguments and explanations in the introduction (Please see line 63-99 in the revised version).
Comment 2: Very poor presentation of results. One example is in lines 43 to 51. What two results are being compared? Correct this and elsewhere in the entire manuscript.
Authors'responses and locations of the revisions: Thanks. Owing to your suggestion, we modified the relevant description of the results. (Please see line 32-53 in the revised version).
Comment 3: Line 11: Dietary fiber is not a nutrient. Please modify the statement.
Authors'responses and locations of the revisions: Thanks. Owing to your suggestion, we modified the sentence as “Dietary crude fiber is an important substance and solid-state fermentation can improve the nutritional value of feed.” (Please see line 11 in the revised version).
Comment 4: Lines 14-17: The treatments abbreviations are ambiguous and very difficult for readers to follow. Instead used the following abbreviations for the treatments: (1) pigs fed a basal diet containing 7.00% CF (HF) instead of (HN); (2) pigs fed a basal multi-bacteria fermentation diet containing 7.00% CF (HFM) instead of (HF); (3) pigs fed a basal diet containing 2.52% CF (LF) instead of (LN); (4) piglets fed a basal multi-bacteria fermentation diet containing 2.52% CF (LFM) instead of (LF)
Authors'responses and locations of the revisions: Thanks. Owing to your suggestion, we repaced the treatments abbreviations “HN” with “HF”, replaced “HF” with “HFM”, replaced “LN” with “LF”, and replaced “LF” with “LFM” in the entire article.
Comment 5: Line 17: There is nothing called factorial design but there is factorial arrangement of the treatment structure.
Authors'responses and locations of the revisions: We have corrected as “a 2 × 2 design”.
Comment 6: Line 28: Do not start sentences with figure. You could start the sentence like this: A total of 36 pigs…………. Correct this in the entire manuscript.
Authors'responses and locations of the revisions: Thanks. Owing to your suggestion, we modified the sentence as “...... a total of 36 finishing pigs (Landrace × Large White × Duroc) with similar initial body weight (70.80 ± 5.75 kg) were randomly divided into 4 treatments with 9 barrows in each ......”, and other sentences like this were modified too (Please see line 13, 28, 140, 165 and 206 in the revised version).
Comment 7: Line 29: Pig numbers in line 29 contradicts that of line 28.
Authors'responses and locations of the revisions: Thanks. Owing to your suggestion, we corrected the pig numbers. The sentence was modified as “...... a total of 36 finishing pigs (Landrace × Large White × Duroc) with similar initial body weight (70.80 ± 5.75 kg) were randomly divided into 4 treatments with 9 barrows in each ......”, and other mistakes like this were corrected too (Please see line 14, 29 and 142 in the revised version ).
Comment 8: Lines 32-43. This sentence is too long. Break it into 3 or 4 sentences. Correct this long sentences in the entire manuscript.
Authors'responses and locations of the revisions: Thanks. Owing to your suggestion, we broke up the long sentences.
The modified sentences as following: Multi-bacteria solid-state fermented diet increased the backfat thickness (P < 0.05) and apparent total tract nutrient digestibility (ATTD) of CF, neutral detergent fiber (NDF), acid detergent fiber (ADF), crude protein (CP), 8 amino acids (Trp, Asp, Gly, Cys, Val, Met, Ile and Leu), total essential amino acids (EAA), total non-essential amino acids (NEEA) and total amino acids (TAA) (P < 0.05). Multi-bacteria solid-state fermented diet increased serum concentrations of HDL-c, TP, ABL and GLU, and serum enzyme activities of T-AOC, GSH-Px, SOD and CAT (P < 0.05), the relative abundance of Lactobacillus, Oscillospira, and Coprococcus (P < 0.05), and the abundance of YAMINSYN3-PWY, PWY-7013, GOLPDLCAT-PWY, ARGORNPROST-PWY, and PWY-5022 pathways (P < 0.05) as well. Multi-bacteria solid-state fermented diet reduced the digestion amount of CF, NDF and ADF (P < 0.05), serum concentrations of TC, TG, LDL-c, BUN, and MDA (P < 0.05), the relative abundance of Streptococcaceae (P < 0.05), and the abundance of PWY-6470, PWY0-862, HSERMETANA-PWY, LACTOSECAT-PWY, MET-SAM-PWY, PWY-6700, PWY-5347, PWY0-1061, and LACTOSECAT-PWY pathways (P < 0.05). High-fiber diet increased average daily feed intake (P < 0.05), serum concentrations of TC, TG, LDL-c, BUN, and MDA (P < 0.05), the relative abundance of Clostridiaceae_Clostridium and Coprococcus (P < 0.05), and the abundance of TCA-GLYOX-BYPASS, GLYCOLYSIS-TCA-GLYOX-BYPASS, and PWY-6906 pathways (P < 0.05). High-fiber diet reduced chest circumference (P < 0.05) and ATTD of ether extract (EE), CF, NDF, ADF, Ca, CP, 18 amino acids (Trp, Thr, Val, Met, Ile, Leu, Phe, Lys, His, Arg Asp, Ser, Glu, Gly, Ala, Cys, Tyr, and Pro), EAA, NEAA and TAA (P < 0.05). High-fiber diet also reduced serum concentrations of HDL-c, TP, ABL, and GLU, and serum enzyme activities of T-AOC, GSH-Px, SOD, and CAT (P < 0.05), and the relative abundance of Akkermansia, Oscillospira (P < 0.05). (Please see line 32-53 in the revised version)
Comment 9: Lines 53: ATTD of what?
Authors'responses and locations of the revisions: Thanks. Owing to your suggestion, we corrected the pig numbers. The sentence was modified as “7.00% CF had a negative effect on digestion of nutrients, but multi-bacteria solid-state fermentation diets could relieve this negative effect and increase backfat thickness.” (Please see line 55-57 in the revised version ).
Comment 10: Lines 70-71: Is it fermentation of microorganisms or fermentation by microorganisms?
Authors'responses and locations of the revisions: Thanks. Owing to your suggestion, we examined it carefully and rewrote the sentence as “Solid-state fermentation refers to the fermentation by microorganisms in a solid substrate with little free water.” (Please see line 76-77 in the revised version).
Comment 11: Lines 104-109: what is the coefficient of variation (CV) of the biochemical analysis?
Authors'responses and locations of the revisions: Thanks. Owing to your suggestion, we added the corresponding coefficient of variation after each biochemical kit
The modified sentences as following: Detection kits of serum total cholesterol (TC) (CV ≤ 3%), total triglycerides (TG) (CV ≤ 5%), high-density lipoprotein (HDL-c) (CV ≤ 3%), low-density lipoprotein (LDL-c) (CV ≤ 8%), total protein (TP) (CV = 1.03%), albumin (ALB) (CV = 2.3%), urea nitrogen (BUN) (CV ≤ 5%), glucose (GLU) (CV ≤ 5%), total antioxidant capacity (T-AOC) (CV ≤ 3.6%), glutathione peroxidase (GSH-Px) (CV v 3.56%), superoxide dismutase (SOD) (CV ≤ 1.7%), catalase (CAT) (CV ≤ 1.7%) and malondialdehyde (MDA) (CV = 1.5%) were purchased from Nanjing Jiancheng Bioengineering Institute (Nanjing, China). (Please see line 116-123 in the revised version).
Comment 12: Lines 132: Delete the phrase “according to” and replace with “to meet or exceed”
Authors'responses and locations of the revisions: Thanks. Owing to your suggestion, we examined it carefully and rewrote the sentence as “The ingredients and the compositions of the basal diet are formulated to meet or exceed the NRC (2012) recommendations (Table 1).” (Please see line 145-147 in the revised version).
Comment 13: Lines 133: The nutritional composition information in table 1, is it analyzed values or calculated values?
Authors'responses and locations of the revisions: The nutritional composition information in table 1 was analyzed values.
Comment 14: Lines 147 to 149: These body measurements were taken at day 28 why not on day 0 as well. The difference that were observed on day 28 could have also existed at day 0. Therefore, there will a need for covariate analysis.
Authors'responses and locations of the revisions: there was no difference on initial body weight on day 0 (please see table 2).
Comment 15: Lines 474: Did authors investigate any metabolic pathway? This is over generalization.
Authors'responses and locations of the revisions: Yes, please see Table 7.
Reviewer 2 Report
Dear authors
Many thanks for this piece of work which I tfind very interesting. I read it with interest and I believe that it would be meritorious of publication after revision.
One main limitation is language style. It needs extensive revision.
Another main limitation is the concept of Crude fibre (which has nutritionally speaking not that meaning authors mean, unless in the case of which fibrous fractions are involved). I would suggest to explore and refere to the paper by Cappai et al., 2013 Food and function.
Results are well.displayed but discussion shoud focus more on the nutritional meaning in my opinion.
So conclusion should stick close to results.
The solid state fermentation with those bacterial species is a valuable intuition.
Author Response
Section B: "Responses to the Comments by reviewer 2"
Comment 1: One main limitation is language style. It needs extensive revision.
Authors'responses and locations of the revisions: Thank you very much for your suggestions, I revised my English style, corrected some errors and improved relevant statements based on your comments.
Comment 2: Another main limitation is the concept of Crude fibre (which has nutritionally speaking not that meaning authors mean, unless in the case of which fibrous fractions are involved). I would suggest to explore and refere to the paper by Cappai et al., 2013 Food and function.
Authors'responses and locations of the revisions: The reference (doi: 10.1039/c3fo60075k) you recommended has been of great help to me, and I improved my argument on this basis.
Comment 3: Results are well.displayed but discussion should focus more on the nutritional meaning in my opinion. So conclusion should stick close to results.
Authors'responses and locations of the revisions: We have revised the “discussion”.
Reviewer 3 Report
Dietary crude fiber is an important nutrient and solid-state fermentation can improve the nutritional value of feed. This article studies the effects of dietary crude fiber levels and multi-bacterial solid-state fermented feeds on finishing pigs, including growth performance, digestibility, intestinal microbial ecology and some metabolites. The results show that High-fiber diets and multi-bacteria solid-state fermentation improved the diversity and abundance of fecal microorganisms in finishing pigs. The design of the paper is reasonable, the results are detailed, the discussion is well organized. The data provide basal reliance reference for growing pigs diets. Evaluating positively the results obtained, however, it is necessary to point out a number of drawbacks:
Simple Summary
Shorten this part.
Line 18: “nutrients” should be “nutrient”
Abstract
Line 40 – Please change “ , and ” to “, and “.
Material and Methods
Line 112 – Please change “ Malondiadehyde “ to “ malondiadehyde “.
Line 214 – Please pay attention to the correct writing of titanium dioxide (TiO2).
Results
Table 4 – Keep the same number of decimal places for all P - values.
Table 5 – Please unify the representation of extremely significant P - values (P< 0.01 or p < 0.001).
Table 7- wrong data show: “0..95”
Author Response
Section C: "Responses to the Comments by reviewer 3"
Comment 1: Simple Summary: Shorten this part
Authors'responses and locations of the revisions: Thanks. Owing to your suggestion, we examined it carefully and rewrote this paragraph.
The modified sentences as following: Dietary crude fiber is an important substance and solid-state fermentation can improve the nutritional value of feed. To study the effects of multi-bacteria solid-state fermented diets and dietary crude fiber levels on finishing pigs, a total of 36 pigs (70.80 ± 5.75 kg) were randomly divided into 4 treatments with 9 barrows in each: (1) pigs fed a basal diet containing 7.00% CF (HF); (2) pigs fed a basal multi-bacteria fermentation diet containing 7.00% CF (HFM); (3) pigs fed a basal diet containing 2.52% CF (LF); (4) piglets fed a basal multi-bacteria fermentation diet containing 2.52% CF (LFM). And the growth performance, nutrient digestibility and digestion amount, serum biochemical index, and fecal microflora were evaluated. Multi-bacteria solid-state fermentation had a positive effect on the nutrients digestion and serum biochemical indicators, which was contrary to high-fiber diets. Both high-fiber diet and multi-bacteria solid-state fermentation could optimize intestinal flora in finishing pigs. (Please see line 11-21 in the revised version).
Comment 2: Line 18: “nutrients” should be “nutrient”
Authors'responses and locations of the revisions: Thanks. Owing to your suggestion, we replaced “nutrients” with “nutrient”. (Please see line 17 in the revised version)
Comment 3: Line 40: Please change “, and ” to “, and ”.
Authors'responses and locations of the revisions: Thanks. Owing to your suggestion, we modified the sentence as “...... the relative abundance of Lactobacillus, Oscillospira, and Coprococcus (P < 0.05), and the abundance of YAMINSYN3-PWY, ......” (Please see line 39 in the revised version)
Comment 4: Line 112: Please change “Malondiadehyde” to “malondiadehyde”.
Authors'responses and locations of the revisions: Thanks. Owing to your suggestion,we replaced “Malondiadehyde” with “malondiadehyde”. (Please see line 121 in the revised version)
Comment 5: Line 214: Please pay attention to the correct writing of titanium dioxide (TiO2)
Authors'responses and locations of the revisions: Thanks. Owing to your suggestion, we replaced “TiO2” with “TiO2”. (Please see line 216 and 225 in the revised version)
Comment 6: Table 4: Keep the same number of decimal places for all P-values.
Authors'responses and locations of the revisions: Thanks. Owing to your suggestion, we examined it carefully and unified the decimal places of all P-values in this table. (Please see Table 4 in the revised version).
Comment 7: Table 5: Please unify the representation of extremely significant P-values (P < 0.01 or P < 0.001).
Authors'responses and locations of the revisions: Thanks. Owing to your suggestion, we examined it carefully and unified the representation of extremely significant P-values as P < 0.001 in this table. (Please see Table 5 in the revised version).
Comment 8: Table 7: wrong data show: “0..95”
Authors'responses and locations of the revisions: Thanks. Owing to your suggestion, we replaced “0..095” with “0.095”. (Please see Table 7 in the revised version).
Round 2
Reviewer 1 Report
Overall, authors effectively responded to the the suggestions, comments, and the questions raised.